# Ictal Bradycardia and Asystole in Sleep-Related Hypermotor Epilepsy: A Study of 200 Patients

**DOI:** 10.3390/jcm13061767

**Published:** 2024-03-19

**Authors:** Lorenzo Muccioli, Giulia Bruschi, Lorenzo Ferri, Anna Scarabello, Lisa Taruffi, Lidia Di Vito, Barbara Mostacci, Federica Provini, Giovanna Calandra-Buonaura, Paolo Tinuper, Laura Licchetta, Francesca Bisulli

**Affiliations:** 1Department of Biomedical and Neuromotor Sciences, University of Bologna, 40139 Bologna, Italy; lorenzo.muccioli@gmail.com (L.M.); lorenzo_ferri@rocketmail.com (L.F.); annascarabello94@gmail.com (A.S.); federica.provini@unibo.it (F.P.); giovanna.calandra@unibo.it (G.C.-B.); paolo.tinuper@unibo.it (P.T.); 2IRCCS Istituto delle Scienze Neurologiche di Bologna, ERN EpiCARE, 40139 Bologna, Italy; bruschigiulia239@gmail.com (G.B.); lisa.taruffi@studio.unibo.it (L.T.); lidia3@gmail.com (L.D.V.); barbara.mostacci@isnb.it (B.M.); laura.licchetta@ausl.bologna.it (L.L.); 3Department of Neuroscience, University of Padua, 35122 Padua, Italy; 4Department of Biomedical, Metabolic and Neural Sciences, Center for Neuroscience and Neurotechnology, University of Modena and Reggio Emilia, 41121 Modena, Italy

**Keywords:** arrhythmia, heart, seizure, polygraphy, focal cortical dysplasia (FCD), MRI, genetics, GATOR1, DEPDC5, SUDEP

## Abstract

**Background**: Ictal bradycardia (IB) and asystole (IA) represent a rare but potentially harmful feature of epileptic seizures. The aim of this study was to study IB/IA in patients with sleep-related hypermotor epilepsy (SHE). **Methods**: We retrospectively included cases with video-EEG-confirmed SHE who attended our Institute up to January 2021. We reviewed the ictal polysomnography recordings focusing on ECG and identified cases with IB (R-R interval ≥ 2 s or a ≥10% decrease of baseline heart rate) and IA (R-R interval ≥ 4 s). **Results**: We included 200 patients (123 males, 61.5%), with a mean age of 42 ± 16 years. Twenty patients (20%) had focal cortical dysplasia (FCD) on brain MRI. Eighteen (out of 104 tested, 17.3%) carried pathogenic variants (mTOR pathway, *n* = 10, nAchR subunits, *n* = 4, KCNT1, *n* = 4). We identified IB/IA in four cases (2%): three had IA (mean 10 s) and one had IB. Three patients had FCD (left fronto-insular region, left amygdala, right mid-temporal gyrus) and two had pathogenic variants in *DEPDC5*; both features were more prevalent in patients with IB/IA than those without (*p* = 0.003 and *p* = 0.037, respectively). **Conclusions**: We identified IB/IA in 2% of patients with SHE and showed that this subgroup more frequently had FCD on brain MRI and pathogenic variants in genes related to the mTOR pathway.

## 1. Introduction

Sleep-related hypermotor epilepsy (SHE) is characterized by motor seizures occurring predominantly during sleep [1]. This condition has a prevalence of 1.8/100,000 individuals [2] and may manifest at any age, with a peak during childhood and adolescence [1].

Seizures are abrupt and brief (typically <2 min), and are characterized by hyperkinetic or asymmetric dystonic/tonic motor patterns, usually with vocalization, emotional facial expressions and vegetative signs [1].

The etiology of SHE may be genetic, genetic–structural, or acquired. Genetic causes include pathogenic variants in genes encoding components of the GATOR1 complex, a negative regulator of mTOR (*DEPDC5*, less frequently *NPRL2* or *NPRL3*), in acetylcholine receptor subunit genes (*CHRNA4*, less frequently *CHRNB2* or *CHRNA2*), and in *KCNT1* [3,4,5]. Individuals with pathogenic variants in GATOR1 complex genes may have focal cortical dysplasia (FCD), and have type II in particular (Nobili et al., 2007) [5].

Autonomic manifestations, notably arrhythmias, occur frequently in epileptic seizures. The structures involved in the central control of the vegetative function, i.e., the central autonomic network (CAN), are the insular cortex, the periaqueductal grey, and the central nucleus of the amygdala, as well as brainstem “autonomic” nuclei such as the nucleus of the solitary tract and the nucleus ambiguous, which, respectively, control the sympathetic and parasympathetic system; these modulate the rate of the sinoatrial node (SA), and, consequently, the heart rate itself [6,7,8].

The most common cardiac arrhythmia in epileptic seizures is sinus tachycardia, which may occur in up to 80% of seizures, usually without sequelae [9]. Conversely, ictal bradycardia (IB) and ictal asystole (IA) have a lower prevalence (2% and up to 0.32% of seizures, respectively) [9,10,11], but their clinical consequences may be more severe, potentially leading to syncope and falls [12]. IB/IA is primarily observed in patients with temporal lobe epilepsy (about 80% of cases), followed by frontal lobe epilepsy [10,13], and appears to occur primarily in association with bilateral hemispheric discharges [14]. The association between the temporal and frontal lobes and structures of the CAN may explain the reason for their frequent involvement in ictal cardiac arrhythmias [10].

The mechanisms underlying sudden unexpected death in epilepsy patients (SUDEP), the most significant category of epilepsy-related deaths (up to 18%), are still poorly understood: they are likely to primarily involve post-ictal respiratory/cardiac events, although IB and IA have been considered as potential contributing factors [15,16]. Variants in several genes have been identified as possible high-risk factors in SUDEP, mainly those encoding sodium channels, but also others such as *DEPDC5* [17].

The aim of this study was to elucidate the prevalence of ictal bradyarrhythmia in patients with SHE and to investigate any distinctive feature of this population.

## 2. Methods

### 2.1. Inclusion Criteria and Classification of Ictal Bradycardia and Asystole

We retrospectively included cases with video-EEG-confirmed SHE [1] who attended our Institute up to January 2021. We collected data regarding demographics, epilepsy history, video-EEG/polysomnography (VPSG), neuroimaging, antiseizure medications, neurosurgery, genetics, cardiologic evaluations and comorbidities.

For each patient, we reviewed the ictal video-EEG recordings focusing on ECG and identified cases with IB/IA. IB was defined as an R-R interval ≥ 2 s [14] or a decrease ≥10% in the baseline heart rate (HR) [18]. IA was defined as an R-R interval ≥ 4 s [19].

### 2.2. Analysis of Seizures with Ictal Bradycardia or Asystole

In patients with seizures arising from sleep, we analyzed episodes that occurred after ≥5 min of seizure-free sleep and ≥25 s after the end of another cortical arousal or transient motor or respiratory event that did not produce awakening, in order to eliminate the confounding effect of potentially persistent HR fluctuations caused by events occurring before the episode under investigation.

HR variations were studied throughout the seizure period in relation to seizure onset (SO).

SO was defined as the first visible discharge on the EEG trace or, in cases where EEG did not change, as the first visible increase in the EMG signal amplitude that coincided with the first clinical manifestation detected by a visual analysis of the video. Seizures in which the first change in EMG signal was preceded by other clinical manifestations (e.g., movements or eye-opening) were excluded.

HR was calculated by measuring the interval between two consecutive R waves (RRi) in the ECG trace. Using Brain Vision Analyzer software (version 2.2), the RRi series were first automatically identified. The semiautomatic function of the software was later used to review the RRi series, allowing the correction of erroneously identified R waves and missed detections. Lastly, R markers were exported and the RRi was calculated for the entire night recorded.

### 2.3. Statistical Analysis

We compared the features of SHE patients with and without IB/IA using Fisher’s exact test for categorical variables and the Mann–Whitney U test (or *t*-test, when appropriate) for continuous variables. Furthermore, the *p*-values calculated comparing the two groups were adjusted by the Benjamini–Hochberg false discovery rate (FDR) multiple testing correction [20]; after the correction, a q-value < 0.10 was considered statistically significant.

## 3. Results

### 3.1. Study Population

We included a total of 200 patients with video-EEG-confirmed SHE, comprising 123 males (61.5%) and 77 females (38.5%). The mean age at the last observation was 42 ± 16 years (range: 3–102 years), while the mean follow-up was 14.9 ± 14.2 years. The clinical features are summarized in Table 1.

The mean age at epilepsy onset was 13.3 ± 10.0 years (range: 0.25–56 years). Most patients presented with one/multiple seizures per night (80, 39.6%), whereas for the remaining cases, the frequency at onset ranged from weekly (35, 17.3%) to sporadic (34, 16.8%). Seventy-four patients (37%) experienced focal to bilateral tonic–clonic seizures. A neuroradiologic assessment was available in all but six patients; 46/194 (23.2%) showed abnormal brain MRI, 20 of whom had a focal lesion consistent with FCD.

Genetic analysis showed a pathogenic variant in SHE-related genes in 18 out of the 104 patients screened (17.3%). In particular, 10 (9.6%) had causative variants in the genes encoding proteins of the mTOR pathway (GATOR-1-related genes, *n* = 9; *TSC2*, *n* = 1), 4 (3.8%) in genes encoding the neuronal nicotinic acetylcholine receptor (nAChR) subunits, and 4 (3.8%) in *KCNT1*. Therefore, of the 200 patients, 8 had a structural and genetic etiology (four related to GATOR1-complex gene), 10 had a genetic etiology, and 38 had an underlying structural lesion. In the remaining 144 cases, the etiology was unknown.

At the last assessment, 147 were taking antiseizure medications, with seizure freedom in 51 patients. Carbamazepine, alone or in combination, had been taken by 71 patients with a medium dose of 677 mg/day; lamotrigine, alone or in combination, had been taken by 15 people, with a mean dose of 325 mg/day.

A total of 51 people (25.5%) underwent a pre-surgical work-up: 16 subsequently underwent epilepsy surgery (mainly frontal corticectomy or lesionectomy), with an excellent surgical outcome (Engel class Ia) in five. Of the remaining 35, 13 were excluded from surgery after the pre-surgical work-up, 10 refused to undergo surgery or stereo-EEG, and 12 were waiting for surgery or stereo-EEG at the time this study was conducted.

### 3.2. Cases with Ictal Bradycardia/Asystole

Following the review of video-EEG recordings, we identified four patients (2%) (M/F = 1/3) with IB or IA. Specifically, three patients (1.5%) (cases 1–3) experienced episodes of IA lasting up to 13 s (mean: 10 s) in the analyzed seizures, while one patient (case 4) with baseline sinus tachycardia had IB, with a HR reduction of 25%. IA/IB was detected in 20–50% of the recorded seizures and appeared after a mean of 10 s from the first motor activity. The features of these patients are summarized in Table 1 and reported below, while details of the analyzed seizures are outlined in Table 2. The example of case 2 is reported in Figure 1.

The mean age at epilepsy onset was 8.1 ± 6.6 (range: 0.25–16 years). The mean age at the last assessment was 41 years (range: 19–60 years).

Seizure frequency at onset was known in two patients, both experiencing one or more seizures per night. The maximum seizure frequency over the lifetime was every night for all these cases. One of them (25%) experienced focal to bilateral tonic–clonic seizures. At the time of analysis, all four patients had seizures occurring exclusively during sleep.

The analysis of ictal or post-ictal semiology did not reveal significant differences between patients with IB/IA and those without IB/IA, neither in terms of severity nor duration.

All patients showed interictal epileptiform abnormalities, bilateral in three (fronto-temporal regions, *n* = 2, fronto-central, *n* = 1) and focal over the left temporal region in the other one.

Brain MRI revealed FCD in three patients (75%). One patient had FCD in the left fronto-insular region, another in the left amygdala, and the last in the right mid-temporal gyrus. Genetic analysis revealed a pathogenic variant in *DEPDC5* in two cases (50%), whereas the other two cases were negative for pathogenic variants in SHE-related genes. Therefore, among the four patients, the etiology was genetic–structural in two, structural in one, and unknown in one.

One patient (25%) had OSA syndrome of moderate–severe entity and required continued positive airway pressure treatment.

At the last assessment, all patients were on ASMs, with seizure control in one (case 2, seizure-free following epilepsy surgery). One patient experienced a >50% reduction in seizures while the other two were non-responders.

All patients underwent a pre-surgical work-up: one patient underwent surgery (case 2, left fronto-insular lesionectomy), two patients (case 3 and 4) are on the list for stereo-EEG at the time of writing, and one patient was excluded from further study (case 1).

Cardiological evaluations, including dynamic echocardiography and baseline ECG recordings, were performed for each patient, revealing no pathological findings and demonstrating normal R-R interval variability.

A cardiac pacemaker (CPM) was implanted in cases 1 and 2, not leading to differences in seizure frequency and semiology.

### 3.3. Clinical Vignettes

*Case 1* (female, 44 years old): at 16 years of age, the patient began experiencing nocturnal seizures characterized by vocalizations and hand and trunk movements that were initially controlled by carbamazepine. At 35 years of age, these seizures recurred and, despite add-on therapy with topiramate, increased up to three seizures/night. VPSG recorded six episodes, lasting up to 96 s, among which three (50%) showed that asystole occurred after a mean of 19 s after the first motor activity and had a mean duration of 17 s (the one available for analysis lasted 11.7 s, see Table 1). Two additional seizures occurred after CPM implantation, pacing the cardiac rhythm for 11 and 39 s. EEG showed a diffuse flattening of background activity, which was then followed by movement artifacts and then no longer interpretable. We documented by VPSG an identical ictal semiology during the seizures recorded before and after implantation, despite the correct CPM activation. Brain MRI showed a lesion consistent with FCD in the left amygdala. Whole-exome sequencing revealed a pathogenic variant (c.1225delA, p.Thr409HisfsTer15) in *DEPDC5*. The patient was excluded from epilepsy surgery workout.

*Case 2* (male, 60 years old): at 10 years of age, the patient presented with multiple seizures per night that were characterized by hyperkinetic movements of the four limbs without impaired awareness. Seizures were refractory to several ASM. He was hospitalized at our institute for the first time at 57 years of age. VPSG captured 11 seizures during light sleep that were characterized by variable intensity and duration (up to 30 s), consisting in the opening of the eyes, grimaces, the sudden lift of the head and trunk, and asymmetric dystonic posture with the predominant involvement of the left side. During two seizures (18%), he presented one episode of IB and one of IA after a mean of 9 s from the first motor activity, lasting up to 13.3 s. After CPM implantation, five further seizures with the same features were recorded, with paced cardiac rhythm in all of them. Interictal EEG revealed left-frontotemporal spikes, while no clear-cut ictal discharge was identified. Brain MRI revealed dysplasia in the left fronto-insular region, whose removal led to seizure freedom after a 2-year follow-up.

*Case 3* (female, 41 years old): at 6 years of age, the patient presented with sleep-related seizures characterized by stiffening and “limb shaking”. The initial frequency was unknown, but in the following years, the seizure frequency increased up to 30 per day despite ASM polytherapy. VPSG captured 39 stereotyped seizures during light sleep, lasting 10–50 s; these were characterized by asymmetric dystonic posture predominantly on the right side, right-limb grasping, automatisms of the left limbs and left unilateral blinking, grimaces, and concomitant impaired awareness. Interictal EEG showed sharp waves in the fronto-central leads bilaterally, while ictal EEG was characterized by low-voltage fast activity in the same regions. Sixteen episodes (41%) were associated with IA at a mean of 16 s after the first motor activity, lasting up to seven seconds. Brain MRI was unremarkable. The patient is on the list for stereo-EEG.

*Case 4* (female, 19 years old): the patient has a family history of focal epilepsy, including SHE, on her father’s side and was born via dystocic delivery for acute fetal distress. At three months of age, she presented with episodes occurring during both sleep and wakefulness that were characterized by a brief stereotypical smile and trismus lasting a few seconds; this was followed by vomiting with unknown frequency and resolved with antiseizure medications. Six months later, she developed sleep-related hypermotor seizures that rarely occurred during wakefulness (last one at 10 years of age). Three episodes were recorded during VPSG during sleep, lasting 42–69 s; these were characterized by early asymmetric bilateral tonic posture predominant on the left side with bilateral tonic–vibratory evolution and guttural noises. Ictal EEG showed discharge on the right fronto-central region with bilateral diffusion, which was then covered by motor artifacts. In one seizure (33%), 5 s before the first motor activity, ECG revealed a slowing from 103 to 75 bpm, i.e., a reduction > 25% from the baseline frequency (which was consistently >100 bpm in prolonged ECG recordings), lasting 14 s. Brain MRI showed a lesion consistent with FCD in the right mid-temporal gyrus. Genetic analysis revealed a pathogenic variant in *DEPDC5* (c.985delA, p.Thr329LeufsTer7), segregating with a 44% penetrance in the family. The patient is on the list for stereo-EEG.

### 3.4. Statistical Comparison of the Two Groups

The variables depicted in Table 1 were compared between the group of SHE patients with IB/IA (*n* = 4) and those without IB/IA (*n* = 196).

The group with IB/IA was found to be associated with abnormal neuroimaging (75% vs. 22.4%, *p* = 0.041), the presence of FCD in brain MRI (75% vs. 8.7%, *p* = 0.003), and pathogenic variants in the mTOR pathway (50% vs. 8%, *p* = 0.045) or GATOR-1 genes (50% vs. 7%, *p* = 0.037), compared with the group without IB/IA. However, after collecting multiple comparisons with FDR, only the presence of FCD remained statistically significantly associated with IB/IA (q = 0.081) compared with the group without IB/IA.

## 4. Discussion

In this study, we included 200 patients with confirmed SHE and identified IB/IA in four cases (2%) (IA, *n* = 3, 1.5%; IB, *n* = 1, 0.5%), described the features of IB/IA and showed that this subgroup of patients more frequently had FCD on brain MRI and pathogenic variants in genes related to the mTOR pathway.

### 4.1. Prevalence of IB/IA

The frequency of IB/IA detected in our cohort was higher compared with the previous literature, which showed a prevalence of IA in 0.3% of epilepsy patients monitored through video-EEG [9] and of IB/IA in less than 2% of persons with epilepsy in large population studies [10,11]. This finding may be attributed to the systematic application of simultaneous EEG/ECG video monitoring, although this has become common practice in recent years, and to the fact that patients with SHE, unlike other epilepsy syndromes, may have several seizures per night, thereby increasing the number of seizures screened and probability of capturing IB/IA, which do not necessarily occur in all seizures [21]. Clearly, the elevated frequency of IA/IB might also stem from the anatomo-electro-clinical features and genetic background of SHE.

### 4.2. Seizure Onset Zone in Patients with IA/IB

In the three cases with a structural etiology, FCD was located in the extra-frontal regions, notably the left insula, right mid-temporal gyrus, and left amygdala, in line with the literature data showing that IB and IA mainly occur in temporal lobe epilepsy (80–82%) and, to a lesser extent, are associated with a frontal (6–10%) or insular (3–5%) origin [13]. The variable seizure onset zone and propagation account for the broad range of latency between the beginning of seizures and the asystole [22]. Importantly, these areas are either part of (insula, amygdala) or closely interconnected (temporal cortex) with the CAN, which is involved in the regulation of cardiac function. It would be beneficial to compare the occurrence and characteristics of IB/IA in patients with SHE to those with temporal lobe epilepsy lacking hypermotor manifestations. The characterization of our patients revealed that half of them had no obvious localization of ictal discharge, one had bilateral ictal activity and one had lateralized right-ictal discharge. Although past research has supported the idea that the intracortical stimulation of the left insular cortex produces IB while the stimulation of the right insular cortex leads to ictal tachycardia [8,23,24,25], the lateralization of cardiac regulation in the right hemisphere for sympathetic control and in the left hemisphere for parasympathetic control was not confirmed by the most recent literature [9,26,27,28]. Whichever the side, the intracortical stimulations indicated a posterior predominance of sympathetic control in the insula, whereas the parasympathetic control seemed more anterior [26].

### 4.3. The Role of Structural and Genetic Factors

The presence of FCD was higher in patients with IB/IA compared to those without (75% vs. 9%), and this association remained significant after correction for multiple comparisons. As previously reported, when FCDs are located in the temporal and insular cortices, they might manifest with IA/IB [29,30]. However, although several structural alterations (hippocampal sclerosis/atrophy, neoplasms, developmental abnormality, post-traumatic lesions, cavernomas, and infectious encephalitis) have been reported in epilepsy patients with IA/IB [13,31], large population studies have not identified a clear correlation between neuroimaging findings and IA/IB [13]. Therefore, it remains unknown whether FCD, regardless of its location, could enhance the risk of IA/IB.

A significant proportion of FCDs are related to genetic factors, especially pathogenic variants in GATOR1-related genes [32]. Notably, in our study, we found a higher proportion of GATOR1 pathogenic variants in the subgroup of patients with IA/IB compared with those without bradyarrhythmia (50% vs. 4%); however, this was not significant after correction for multiple comparisons. This higher prevalence is consistent with the higher prevalence of FCD in the former group. Several clinical observations suggest that mutations in GATOR1-related genes confer a higher risk of SUDEP; this has been reported in up to 10% of affected families [33]. *DEPDC5* is expressed in both the brain and heart; however, neither structural nor functional cardiac damage was associated with *DEPDC5* mutations. In *Depdc5c* heterozygous knockout mice, fatal seizures were mediated by brainstem spreading depolarization, with subsequent cardiorespiratory arrest, rather than by cardiac arrhythmia [33]. However, the exact pathophysiological mechanisms underlying SUDEP in patients with GATOR-1-related mutations are still unknown. Similarly, it is not clear whether these genetic factors may have a direct role in facilitating IB/IA—through mechanisms not known thus far—or if this is mediated by their relationship with FCD and the known risk factors for SUDEP (early age at epilepsy onset, drug resistance, sleep-related seizures). It should be kept in mind that the incidence of SUDEP in SHE patients does not differ from that observed in the general epilepsy population (0.36 per 1000 person-years) [34], possibly due to the low prevalence of tonic–clonic seizures.

### 4.4. Semiology and Management of IB/IA

While arrhythmias following T-C seizures are associated with a high risk of SUDEP, ictal arrhythmias are considered of benign nature [9]. It has been suggested that asystole-induced hypoperfusion can even shorten the duration of seizures [22,35]. However, IA can lead to seizure-induced syncope and related traumatic falls, impacting individuals’ quality of life [12,36].

Guidelines for managing individuals with IA/IB are currently lacking. In patients having uncontrolled seizures with IA/IB-related syncopes despite ASM therapy and epilepsy surgery (or in whom this is not feasible), CPM has been shown to reduce morbidity related to falls [37]. On the contrary, since no link between IA and SUDEP has been proven, IB/IA without syncope may not require CPM implantation [36], but this decision should be taken on an individual basis. CPM does not influence seizure frequency but usually alters seizure semiology, avoiding manifestations of cerebral anoxia.

Ictal and post-ictal semiology did not reveal differences between patients with and without IB/IA, yet the limited number of IB/IA cases may have hindered our ability to draw meaningful comparisons.

Notably, despite the relatively high duration of IA/IB in our cohort, none of our patients experienced syncope or presyncope, and motor phenomena persisted throughout the episodes. While it is true that syncopal episodes are less common in patients experiencing seizures during sleep due to their lying position, it is important to note that syncopal episodes can also occur in the supine position, particularly in patients with bradycardia and asystole [38]. Case 1 and 2 underwent CPM implantation due to frequent and prolonged episodes of IA, yet the ictal semiology did not change in terms of duration and ictal motor sequences, compared with seizures with prolonged asystole without CPM activation. These findings emphasize the role of video-polysomnography in the accurate diagnosis and clinical description of ictal features of paroxysmal events of any nature.

### 4.5. Limitations

We acknowledge that the number of cases with IB/IA in our study is limited, and it is indeed challenging to draw general conclusions based on such a small sample size. However, it is important to note that IA/IB and SHE are rare conditions, and our study represents the largest case series in which this phenomenon has been specifically investigated. We believe that highlighting these preliminary results can provide valuable insights and serve as a basis for future research in this area, yet it is crucial to emphasize that our findings are not automatically generalizable to the entire population of patients with SHE. Furthermore, we conducted statistical analyses, including the Benjamini–Hochberg method, which confirmed the positive association with FCD even with this limited sample size. This aspect underscores the robustness of our results despite the small number of cases.

## 5. Conclusions

Our study identified IB/IA in 2% of patients with SHE and showed that this subgroup of patients more frequently had FCD on brain MRI and a trend towards more frequent pathogenic variants in genes related to the mTOR pathway. IB/IA may be a distinct phenotype within the spectrum of SHE, which is associated with specific structural and genetic factors. Further studies are needed to confirm our findings and explore the underlying mechanisms of IB/IA in SHE.

## Figures and Tables

**Figure 1 jcm-13-01767-f001:**
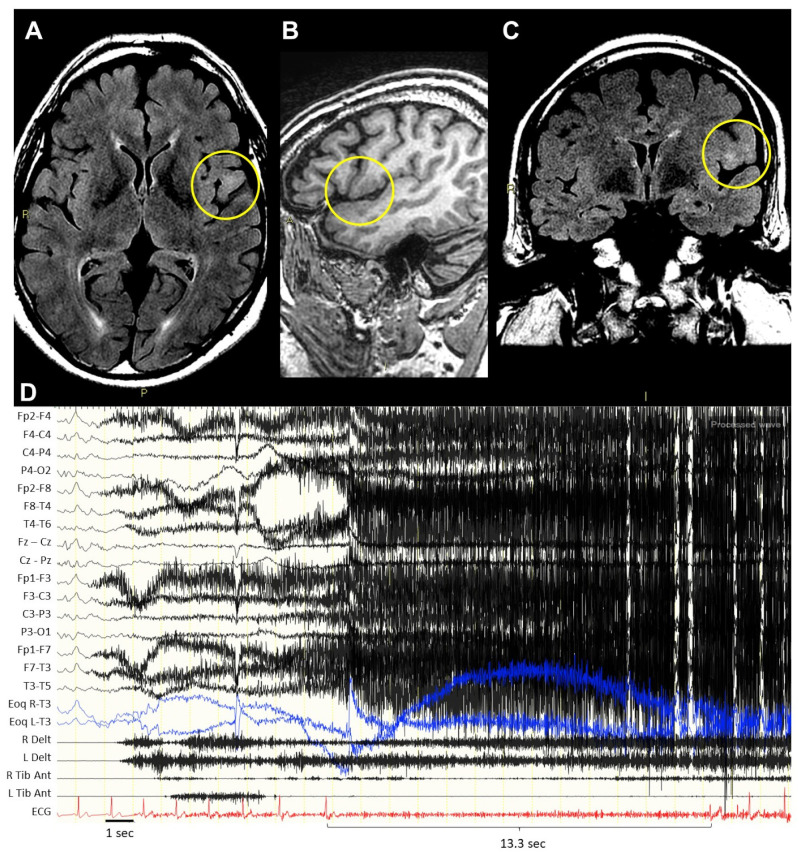
Neuroradiological and neurophysiological features of case 2. Brain MRI ((**A**) axial, FLAIR; (**B**), sagittal, T1; (**C**), coronal, FLAIR) showing an area consistent with focal cortical dysplasia in the left fronto-insular region (yellow circle). Ictal EEG-polygraphy: (**D**) EEG shows a diffuse slow wave, then it is covered by muscle artifacts; ECG shows bradycardia followed by asystole, lasting 13.3 s.

**Table 1 jcm-13-01767-t001:** Features of the study population.

	Total Cohort (*n* = 200)	Cases with Ictal Bradycardia or Asystole			
		Case 1	Case 2	Case 3	Case 4
Age at last follow-up	42 ± 16 y ^a^	44 y	60 y	41 y	19 y
Sex	77 F (38.5%)	F	M	F	F
Age at epilepsy onset	13 ± 10 y ^a^	16 y	10 y	6 y	0.25 y
Seizures frequency at onset		Daily	Multiple/day	NA	NA
- Daily or multiple/day	80 (40%)				
- Weekly or multiple/week	35 (17.5%)				
- Monthly or multiple/month	31 (15.5%)				
- Yearly or multiple/year	34 (17%)				
- Not available	20 (10%)				
Seizures in wakefulness	109 (54.5%)	No	No	No	Yes
Aura	100 (50%)	No	No	Yes	Yes
Focal to bilateral tonic-clonic seizures	74 (37%)	No	No	Yes (once)	No
Status epilepticus	20 (10%)	No	No	No	Yes
Epileptiform interictal EEG	121 (60.5%)	Yes, bilateral fronto-temporal and vertex	Yes, left fronto-temporal	Yes, bilateral fronto-central and vertex	Yes, bilateral fronto-centro-temporal
Ictal paroxysmal activity	115 (57.5%)	Yes, diffuse without observable focal onset	No	Yes, bilateral fronto-central and vertex	Yes, right fronto-centro-temporal
Pathological neurological exam	15 (7.5%)	No	No	No	No
Neuroimaging abnormalities	46 (23%) 20 FCD (10%)	FCD, left amygdala	FCD, left fronto-insular	No	FCD, right mid temporal
Pathogenic variant in SHE-related genes	18	Yes (DEPDC5)	No	No	Yes (DEPDC5)
- mTOR pathway	10				
- Ach receptors	4				
- KCNT1	4				
Personal history					
- Febrile seizures	14 (7%)	No	No	No	No
- Perinatal insult	10 (5%)	No	No	No	Yes
- Intellectual disability/borderline IQ	25 (12.5%)	No	No	No	No
- Developmental delay	11 (5.5%)	No	No	No	Yes
- Psychiatric disorders	45 (22.5%)	No	No	No	Yes
- Obstructive sleep apnea syndrome	18 (9%)	No	Yes	No	No
Family history					
- Febrile seizures	22 (11%)	No	No	Yes	No
- Epilepsy (total)	35 (17.5%)	No	No	Yes	Yes
- Epilepsy (SHE)	15 (7.5%)	No	No	No	Yes
- NREM parasomnias	68 (34%)	No	Yes	No	No
- REM parasomnias	24 (12%)	No	No	No	No
- Intellectual disability	21 (10.5%)	Yes	No	No	Yes
- Psychiatric disorders	32 (16%)	No	Yes	No	No

^a^ mean ± standard deviation; Abbreviations: FCD, focal cortical dysplasia; NA, not available; SHE, sleep-related hypermotor epilepsy.

**Table 2 jcm-13-01767-t002:** Analyzed seizures with ictal bradycardia/asystole.

	Baseline HR	Ictal Behavior (Video)	EEG Seizure Onset	Time to IB	IB Duration	Ictal HR (min)	R-Ri (Max)
**Case 1**	63.7 bpm	Vocalizations, repeated beating of the right hand on the bed, flexion of the head and trunk forward as if trying to get up, stiffening of the right limbs, bimanual and bipedual automatisms. Urine loss was associated.	Diffuse flattening of background activity	28.0 s	14 s	5.1 bpm	11.7 s
**Case 2**	54.3 bpm	Eyes opening, grimaces, sudden lift of the head and trunk, asymmetric dystonic posture with elevation of the lower limbs (left > right).	No identifiable EEG discharge (diffuse slow-wave at onset)	7.7 s	19.6 s	4.5 bpm	13.3 s
**Case 3**	58.7 bpm	Asymmetric dystonic posture (predominant on the right side), grasping with the right upper limb, repetitive purposeless movements of the left limbs, left unilateral blinking, grimaces, and impaired awareness	Low-voltage fast activity in the vertex and bilateral fronto-central regions	18.5 s	10.6 s	9.3 bpm	6.4 s
**Case 4**	103.0 bpm	Early asymmetric bilateral tonic posture (predominant on the left side), with bilateral tonic–vibratory evolution and guttural noises.	Right fronto-centro-temporal discharge and bilateral diffusion	5.4 s before seizure onset	14 s	75.2 bpm	0.8 s

Abbreviations: bpm, beats per minute; HR, heart rate; IB, ictal bradycardia; time to IB, latency between electrographic/motor seizure onset and onset of bradycardia; IB duration, duration of bradycardia; R-Ri, interval between two subsequent R waves on ECG.

## Data Availability

The data presented in this study are available upon request from the corresponding author.

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
