# Peer review of "Ictal Bradycardia and Asystole in Sleep-Related Hypermotor Epilepsy: A Study of 200 Patients"

_jcm, 2024, doi:10.3390/jcm13061767_

Round 1

Reviewer 1 Report

Comments and Suggestions for Authors

I congratulate the authors for their excellent work

The number of patients studied is very high.

The take-home message is as simple as it is important: the role of video-polysomnography in the accurate diagnosis and clinical description of ictal features of paroxysmal events of any nature,  the importance of a genetic evaluation and an accurate neuroimaging study in patients with sleep-related hypermotor epilepsy.

Author Response

Comment: I congratulate the authors for their excellent work. The number of patients studied is very high. The take-home message is as simple as it is important: the role of video-polysomnography in the accurate diagnosis and clinical description of ictal features of paroxysmal events of any nature,  the importance of a genetic evaluation and an accurate neuroimaging study in patients with sleep-related hypermotor epilepsy.

Response: We express our gratitude for your thoughtful feedback, and we greatly appreciate the time and effort you dedicated to reviewing our manuscript.

Reviewer 2 Report

Comments and Suggestions for Authors

Ictal bradycardia and asystole/SHE

This is a single center, retrospective, case series on the prevalence of ictal heart rate slowing in frontal-lobe, sleep-associated epilepsies (SHE). 

1) The manuscript as available to download was incomplete and lacked Tables and Figure. Therefore, my review is incomplete until these are available. The remainder of my review will focus on available information. 

2) Methods/Results: please define the duration required to qualify for IB or IA. As written, it appears that a single heartbeat interval of >2 seconds qualifies. Did the duration or severity of IB/IA (ie, duration) correlate with any particular feature of the patient or epilepsy? I see that Results indicated that indeed very transient HR slowing was included as IB.

3) SHE seizures are typically short without postictal sequelae. How did  the seizures of the 4 IB/IA patients compare to the median/mean of non IB/IA patients?

4) The data shows that SHE had a low prevalence/incidence of IB/IA, and it appears that, perhaps, the incidence of cardiac slowing is higher in similar studies of TLE. I think the authors have the opportunity and techniques to compare their SHE findings to TLE, a consideration for future work that they may like to indicate. 

5) Could the authors comment on their statement in Discussion that “despite the relatively high duration of IA/IB in our cohort, none of our patients experienced syncope” which seems likely given that all were having their seizures while in bed asleep? Please confirm that indeed their selection criteria included only patients/seizures that occurred during sleep? If this assumption was not present, please report on the proportions of awake vs asleep seizures in the sample as well as in the 4 target patients. 

Author Response

We express our gratitude for your thoughtful feedback, and we greatly appreciate the time and effort you dedicated to reviewing our manuscript.

 Comment#2.1 This is a single center, retrospective, case series on the prevalence of ictal heart rate slowing in frontal-lobe, sleep-associated epilepsies (SHE). The manuscript as available to download was incomplete and lacked Tables and Figure. Therefore, my review is incomplete until these are available. The remainder of my review will focus on available information. 

Response#2.1. We apologize for the incomplete manuscript that was available for downloading, which will be provided in the re-submission.

Comment#2.2 Methods/Results: please define the duration required to qualify for IB or IA. As written, it appears that a single heartbeat interval of >2 seconds qualifies. Did the duration or severity of IB/IA (ie, duration) correlate with any particular feature of the patient or epilepsy? I see that Results indicated that indeed very transient HR slowing was included as IB.

Response#2.2 As we reported in the methods and according to literature (Britton J, 2006; Schuele, S. U., 2007; Moseley, B. D., 2011), IB refers to an R-R interval ≥2 seconds or a reduction of heart-rate >10% from baseline, while IA is defined as R-R interval ≥ 4 seconds. With regard to the shortness of HR slowing, the minimum duration of IA was 6.4 seconds while the minimum duration of IB episode was 14 seconds.

We did not notice distinctive features of patients/epilepsy in relation to IB/IA severity and duration. We modified the text accordingly in the results and discussion.

Comment#2.3 SHE seizures are typically short without postictal sequelae. How did  the seizures of the 4 IB/IA patients compare to the median/mean of non IB/IA patients?

Response#2.3 We did not find significant differences between the two groups of patients, although it may be due to the limited number of patients with IA/IB. We modified the text accordingly in results (The analysis of ictal or post-ictal semiology did not reveal significant differences between patients with IB/IA and those without IB/IA, neither in terms of severity nor duration.) and discussion (Ictal and post-ictal semiology did not reveal differences between patients with and without IB/IA, yet the limited number of IB/IA cases may have hindered our ability to draw meaningful comparison.)

Comment#2.4 The data shows that SHE had a low prevalence/incidence of IB/IA, and it appears that, perhaps, the incidence of cardiac slowing is higher in similar studies of TLE. I think the authors have the opportunity and techniques to compare their SHE findings to TLE, a consideration for future work that they may like to indicate. 

Response#2.4 As indicated in the discussion, we showed that the prevalence of IB/IA in our cohort was actually somewhat higher than that reported in previous studies (The frequency of IB/IA detected in our cohort was higher compared with previous literature, which showed a prevalence of IA in 0.3% in epilepsy patients monitored through video-EEG and of IB/IA in less than 2% of persons with epilepsy in large population studies) but are in line with a higher incidence of IB/IA in patients with an extrafrontal/temporal seizure onset zone (In the three cases with a structural etiology, the FCD was located in extra-frontal regions, notably left insula, right mid temporal gyrus, and left amygdala, in line with literature data showing that IB/IA mainly occur in temporal lobe epilepsy (80-82%) and, to a lesser extent, are associated with a frontal (6-10%) or insular (3-5%) origin).

We agree with the reviewer that it would be a great opportunity to compare the findings of the present study with patients with “classic” TLE, i.e. without hypermotor manifestations. This consideration was added in the discussion.

It would be beneficial to compare the occurrence and characteristics of IB/IA in patients with SHE to those with temporal lobe epilepsy lacking hypermotor manifestations.

Comment#2.5 Could the authors comment on their statement in Discussion that “despite the relatively high duration of IA/IB in our cohort, none of our patients experienced syncope” which seems likely given that all were having their seizures while in bed asleep? Please confirm that indeed their selection criteria included only patients/seizures that occurred during sleep? If this assumption was not present, please report on the proportions of awake vs asleep seizures in the sample as well as in the 4 target patients. 

Response#2.5 We confirm that the four patients experiencing IB/IA at the time of analysis had only seizures occurring during sleep. Only one of them (case 4) had also experienced rarer seizures during wakefulness in the past. In the results section, we specified that "At the time of analysis, all four patients had seizures occurring exclusively during sleep."

The presence of seizures occurring during wakefulness was not an exclusion criterion for the diagnosis of SHE, according to the diagnostic criteria outlined by Tinuper et al. in Neurology 2016.

While it is true that syncopal episodes are less common in patients experiencing seizures during sleep due to their lying position, it is important to note that syncopal episodes can also occur in the supine position, particularly in patients with bradycardia and asystole.

This comment was added in the discussion.

Reference: Walsh K, Hoffmayer K, Hamdan MH. Syncope: diagnosis and management. Curr Probl Cardiol. 2015;40(2):51-86. doi:10.1016/j.cpcardiol.2014.11.001  

Reviewer 3 Report

Comments and Suggestions for Authors

The study is clinically relevant and authors explore an important topic trying to see prevalence of ictal asystole and bradycardia in patients with sleep related hypermotor epilepsy. There were only 4 patients out of 200 which IA/IB and met the criteria, but in my opinion, further conclusions can not be drawn from this finding given the paucity of cases. The association with FCD and region and the relationship with pathogenic variants seems far fetched given only 4 cases.  

In addition to the above mentioned comments, would recommend providing more data on IB/IA. How many seizures were evaluated per patient? and how many seizures had IA/IB?  Mean time of onset of IA/IB from seizure onset. 

Authors also mentioned "This finding may be attributed to the systematic application of simultaneous EEG/ECG video monitoring, which is not routinely performed in all institutes", is this true and backed up by any literature? As we know that respiratory belts are not always employed but almost all EEGs have simultaneous one lead recording of EKG.

Author Response

We express our gratitude for your thoughtful feedback, and we greatly appreciate the time and effort you dedicated to reviewing our manuscript.

Comment#3.1The study is clinically relevant and authors explore an important topic trying to see prevalence of ictal asystole and bradycardia in patients with sleep related hypermotor epilepsy. There were only 4 patients out of 200 which IA/IB and met the criteria, but in my opinion, further conclusions can not be drawn from this finding given the paucity of cases. The association with FCD and region and the relationship with pathogenic variants seems far fetched given only 4 cases.  

Response#3.1 Thank you for your thoughtful review. We acknowledge that the number of cases in our study is limited, and it is indeed challenging to draw general conclusions based on such a small sample size. However, it is important to note that IA/IB and SHE are rare conditions, and our study represents the largest case series where this phenomenon has been specifically investigated.  We believe that highlighting these preliminary results can provide valuable insights and serve as a basis for future research in this area, yet it is crucial to emphasize that our findings are not automatically generalizable to the entire population of patients with SHE. Furthermore, we conducted statistical analyses, including the Benjamini-Hochberg method, which confirmed the positive association with FCD even with this limited sample size. This aspect underscores the robustness of our results despite the small number of cases.

This comment has been added in a new “Limitations” section at the end of the discussion.

Comment#3.2In addition to the above mentioned comments, would recommend providing more data on IB/IA. How many seizures were evaluated per patient? and how many seizures had IA/IB?  Mean time of onset of IA/IB from seizure onset. 

Response#3.2 In the results, we described more extensively the case histories of the four patients and the number of episodes recorded with details on IB/IA.

Specifically: Case 1 “VPSG recorded six episodes, lasting up to 96 seconds, among which three (50%) showed asystole occurring after a mean of 19 seconds after the first motor activity and with a mean duration of 17 seconds (the one available for analysis lasted 11.7 seconds, see Table 1). Two additional seizures occurred after CPM implantation, pacing cardiac rhythm for 11 and 39 seconds.”

Case 2: “VPSG captured 11 seizures from light sleep characterized by variable intensity and duration (up to 30 seconds), consisting in eyes opening, grimaces, sudden lift of the head and trunk, asymmetric dystonic posture with predominant involvement of the left side. During two seizures (18%) he presented one episode of IB and one of IA after a mean of 9 seconds from the first motor activity, lasting up to 13.3 seconds. After CPM implantation, five further seizures with the same features were recorded, with paced cardiac rhythm in all of them.”

Case 3: “VPSG captured 39 stereotyped seizures from light sleep, lasting 10-50 seconds, characterized by asymmetric dystonic posture predominant on the right side, right-limb grasping, automatisms of the left limbs and left unilateral blinking, grimaces, with concomitant impaired awareness. Interictal EEG showed sharp waves in the fronto-central leads bilaterally, while ictal EEG was characterized by low voltage fast activity in the same regions. Sixteen episodes (41%) were associated with IA after a mean of 16 seconds after the first motor activity, lasting up to seven seconds”

Case 4: “Three episodes were recorded during VPSG during sleep, lasting 42-69 seconds, characterized by early asymmetric bilateral tonic posture predominant on the left side with bilateral tonic-vibratory evolution and guttural noises. Ictal EEG showed a discharge on the right fronto-central region with bilateral diffusion and then covered by motor artifacts. In one seizure (33%), 5 seconds before the first motor activity, ECG revealed a slowing from 103 to 75 bpm, i.e. a reduction >25% from baseline frequency (which was consistently > 100 bpm in prolonged ECG recordings), lasting 14 seconds.”

Comment#3.3 Authors also mentioned "This finding may be attributed to the systematic application of simultaneous EEG/ECG video monitoring, which is not routinely performed in all institutes", is this true and backed up by any literature? As we know that respiratory belts are not always employed but almost all EEGs have simultaneous one lead recording of EKG.

Response#3.3 Thank you for your comment. We agree that simultaneous EEG/ECG monitoring has become common practice in recent years, while in the past this was not routinely used in all centers. We modified the text according to your suggestion.
“This finding may be attributed to the systematic application of simultaneous EEG/ECG video monitoring, although this has become common practice in recent years

Reviewer 4 Report

Comments and Suggestions for Authors

The article is focused on the analysis of heart rhythm disturbances in patients with sleep-related hypermotor epilepsy (SHE). However, since the manuscript lacks the tables and figures mentioned in the text, it is not possible to properly assess the significance of the findings.

Author Response

Comment#4.1 The article is focused on the analysis of heart rhythm disturbances in patients with sleep-related hypermotor epilepsy (SHE). However, since the manuscript lacks the tables and figures mentioned in the text, it is not possible to properly assess the significance of the findings.

Response#4.1 We apologize for the incomplete manuscript that was available for downloading, which will be provided in the re-submission.

Round 2

Reviewer 2 Report

Comments and Suggestions for Authors

The authors addressed my critiques appropriately.